# The Inventory of Personality Organization-Reality Testing Subscale and Belief in Science Scale: Confirmatory factor and Rasch analysis of thinking style measures

**Andrew Denovan**[1]⊛*, **Neil Dagnall**[2]⊛, **Ken Drinkwater**[2]⊛, **Álex Escolà-Gascón**[3]⊛

**1** School of Psychology, Liverpool John Moores University, Liverpool, United Kingdom, **2** Department of Psychology, Manchester Metropolitan University, Manchester, United Kingdom, **3** Department of Quantitative Methods and Statistics, Comillas Pontifical University, Madrid, Spain

⊛ These authors contributed equally to this work.
* a.m.denovan@ljmu.ac.uk

**Data Availability Statement:** Data is accessible through figshare: https://figshare.com/s/10f60453edc196547316.

## Abstract

The Inventory of Personality Organization-Reality Testing Subscale (IPO-RT) and Belief in Science Scale (BIS) represent indirect, proxy measures of intuitive-experiential and analytical-rational thinking. However, a limited appraisal of factorial structure exists, and assessment of person-item functioning has not occurred. This study assessed the IPO-RT and BIS using confirmatory factor analysis (CFA) and Rasch analysis with a sample of 1030 participants (465 males, 565 females). Correlation analysis revealed a negative, moderate relationship between the measures. CFA supported a bifactorial model of the IPO-RT with four bifactors (Auditory and Visual Hallucinations, Delusional Thinking, Social Deficits, and Confusion). A one-factor model best fitted the BIS. Satisfactory item/person reliability and unidimensionality was observed for both measures using Rasch analysis, and items generally exhibited gender invariance. However, IPO-RT items were challenging, whereas BIS items were relatively easy to endorse. Overall, results indicated that the IPO-RT and BIS are conceptually sound, indirect indices of intuitive-experiential and analytical-rational thinking. Acknowledging the breadth of these thinking styles, a useful future research focus includes evaluating the performance of IPO-RT and BIS alongside objective tests.

## Introduction

Theorists have frequently employed dual processing models to explain how discrete preferences in thinking style/information processing influence real world problem solving and decision making (e.g., conspiracy endorsement, [1, 2]; perception of risk, [3]; scientifically unsubstantiated beliefs, [4]). While these models vary in terminology, emphasis, and complexity, the dual processing approach provides a useful, cogent framework for conceptualising distinct variations in data evaluation and reasoning [5, 6].

The central tenet of thought-based models is that while two separate systems exist and operate interactively, one based on individual inclination or preference predominates. The key,

**Funding:** The author(s) received no specific funding for this work.

**Competing interests:** The authors have declared that no competing interests exist.

recurring distinction within these models is the presence of cognition and emotion driven systems [7]. The former processes information based on rational appeal and functions analytically, whereas the latter is driven by affective impact/influence and intuition. Commensurate with these delineations, cognitive driven processing is conscious, demanding, and effortful, while emotion driven processing derives from associations, feeling, and general heuristics and is automatic, less demanding, and rapid.

A prominent dual processing model is Cognitive-Experiential Self-Theory (CEST) [8–11]. CEST is significant from both a theoretical and measurement perspective as evidenced by the volume of related publications and citations. In addition to the theoretical contribution of CEST, the model also informed the development of the Rational-Experiential Inventory (REI) [12], which is a widely used self-report measure of processing style. CEST is a theory of personality that combines psychodynamic (emotional unconscious of psychoanalysis), cognitive science (affect-free unconscious), and learning theory principles.

Accordingly, CEST comprises analytical-rational and intuitive-experiential systems. The analytic-rational has a limited capacity, is inferential, intentional, and operates through reason. Hence, it draws heavily on higher-order cognitive functions [13]. Whereas the intuitive-experiential is fast, unconscious, holistic, and emotion-based. Thus, outcomes derive from general principles and prior experience. These systems are connected to the extent that analytic-rational processes can consciously moderate the influence of the intuitive-experiential system. Commensurate with this conceptualisation, analytic-rational processing is considered more evolved and sophisticated. Due to the dominance of one within individuals, theorists view processing preference as a trait-like difference [14].

This conceptualisation is operationalised with the REI, which encompasses subscales evaluating Faith in Intuition (intuitive-experiential) and Need for Cognition (analytic-rational). Since development, due to its theoretical coherence and attested psychometric properties (i.e., acknowledged reliability and validity) the REI has featured prevalently within dual processing research [15]. However, because models differ (e.g., System 1 and System 2 thinking [16]) researchers have also employed a range of alternative self-report instruments to assess the systems. For example, investigators have recently used the Inventory of Personality Organization-Reality Testing Subscale (IPO-RT) [17] to assess proclivity to process intra-psychic material (intuitive-experiential) and the Belief in Science Scale (BIS) [18] to evaluate inclination to engage with external, objective evidence (analytic-rational).

The IPO-RT appraises the ability to monitor social reality and distinguish self from non-self (i.e., external from intra-psychic stimuli) [19]. This includes cognitive, perceptual, affective, and social factors (i.e., Auditory and Visual Hallucinations, Delusional Thinking, Social Deficits, and Confusion) [20, 21]. Thus, the IPO-RT in comparison to other commonly used instruments samples a broader range of intuitive-experiential phenomena. For instance, Faith in Intuition, which is restricted to trust in instincts and intuitions [4]. Extensive assessment of construct domain is advantageous from theoretical and measurement perspectives because it enables researchers to better evaluate which specific aspects of intuitive-experiential processing influence decision making.

The BIS assesses the extent to which individuals value science as a source of superior knowledge and advocate the use of the scientific method. This focus aligns well with Need for Cognition, which is the motivational tendency to engage in effortful thinking [22]. High Need for Cognition manifests as the desire to seek, acquire, reflect, and make sense of information [23]. The benefit of using BIS as an index of preference for analytical-rational processing is that attitudes and appreciation of the values of science have an important influence on real world attitudes and actions.

This was illustrated during the COVID-19 pandemic with regards to adherence to Government guidelines and vaccine hesitancy. Despite this, to date, the use of BIS within studies has been limited (e.g., [24–26]). Nonetheless, the construct has merit because faith in the scientific method embodies the desire to systematically test and objectively validate hypotheses about the world. Moreover, these elements are generally indicative of a preference for critical thinking [18].

Support for the notion that the IPO-RT and BIS assess different thinking and processing preferences was provided by Williams et al. [26], who reported a negative relationship between the two constructs. In addition, Dagnall et al. [24] observed a reduction in IPO-RT scores within respondents who scored above the median on BIS. This suggested, consistent with CEST, that greater emphasis on objectivity reduces the importance of subjectively generated data. IPO-RT and BIS also perform comparably to other related measures. For example, outcomes using the IPO-RT have consistently aligned with preceding research using the Faith in Intuition subscale of the REI. Particularly, higher IPO-RT scores have correlated positively with endorsement of scientifically unsubstantiated beliefs (e.g., paranormal credence, conspiracy advocacy, and validation of urban legends). Similarly, BIS, like other measures of analytical thinking (e.g., the Cognitive Reflection Test), correlates negatively with religiosity [27].

Collectively, these findings support the notion that the IPO-RT and BIS function as satisfactory indirect proxy measures of preferential thinking style. In the context of the scales, this is best demarcated as the tendency to rely on spontaneous, subjective internally generated material vs. systematic, objective appraisal of external data.

## The present study

At a measurement level, the IPO-RT and BIS demonstrate satisfactory psychometric properties (i.e., good internal consistency and convergent validity) [20, 28]. However, further work is required to validate the factorial structure. This is necessary since previous analysis of the IPO-RT has recommended different potential solutions and assessment of the BIS, due to the instrument's newness, remains relatively underdeveloped [3].

In the case of the IPO-RT, preliminary examination indicated unidimensionality [17]. However, Ellison and Levy [29] reported subsequently that IPO-items related to reality testing deficits loaded on more than one factor. Acknowledging this issue, Dagnall et al. [20] further scrutinised the IPO-RT and identified four subfactors: Auditory and Visual Hallucinations, Delusional Thinking, Social Deficits, and Confusion, and an additional general factor. The emergent solution was bifactorial as it specified that the IPO-RT functioned adequately at both global and factorial levels. With reference to BIS, only one study has tested the unidimensional solution proposed by Farias et al. [18]. This found strong evidence for a single factor [24].

A further issue with previous psychometric evaluation of the IPO-RT and BIS is the use of classic test theory (CTT). CTT focuses on validity and reliability. This typically involves assessing scales using factor analysis to establish within-item coherence and measurement consistency both internally and across trials (test-retest) [30]. A major limitation of CTT is the presumption that observed scores, in the absence of measurement error, produce true scores.

Noting this, since measurement instruments are imperfect, error is inevitable. However, this is not problematic because the error is random and distributed equally across individuals [31]. Item response theory (IRT) questions this assumption by contending that variations in item difficultly produce systematic error. Consequently, test scores represent a combination of ability and item difficulty. Hence, individual differences (unobservable latent traits) have a manifest effect on observed responses [31].

Rasch models [32] address this issue by determining expected item responses (dichotomous [32] and polytomous [33]) for metric level measurement [34]. This is accomplished by employing a probabilistic form of Guttman scaling [35] and fit statistics [36] to determine the extent to which observed responses correspond to expected values. A logistic function, representing the relative distance between the item and respondent location on a linear scale, determines the probability of a respondent confirming an item. This ensures that responses reflect ability (latent trait) and item difficulty; higher ability being associated with increased likelihood of a correct response. Commensurate with this approach, the probability of a correct response is 0.5% when latent trait location is equal to item difficulty [37]. Within Rasch analyses, a curve classifies the level at which an item maximally distinguishes ability.

Noting these concerns, objectives of the current study included examining the measurement properties of the IPO-RT and BIS using Rasch scaling in conjunction with classic test theory (i.e., confirmatory factor analysis, CFA). CFA evaluated competing latent models of the IPO-RT (one-factor, four-factor, four-factor bifactor) alongside a one-factor BIS model. Subsequently, Rasch analysis tested measurement features, including rating scale efficacy, reliability, item difficulty, dimensionality, and differential item functioning (Fig 1 includes a flowchart depicting the research phases). Rasch analysis was necessary since researchers have not previously assessed item difficulty. Extreme variations in item responses are problematic. If items are too readily validated then scale scores are inflated, and if they are rarely endorsed totals are reduced. Therefore, scales should only contain items that are productive to measurement. Acknowledging this, without Rasch analysis, poorly performing items may undermine measurement efficacy by distorting relationships between the IPO-RT and BIS and other variables.

Using Rasch scaling in conjunction with CFA, due to its emphasis on unidimensionality and ability to estimate ability and item difficulty, is an advised approach for assessing factorial structure [38]. This in the case of the IPO-RT involved determining the best fit of previously posited solutions (one, four, and bifactor). For BIS, evaluation examined unidimensionality.

A further advantage of Rasch scaling is that it enables analysts to identify bias arising from differential item functioning (DIF) [39]. DIF is problematic because it signifies that group membership (i.e., gender and age) is affecting responses. Thus, items are not accurately assessing latent trait or ability level and display a different response probability [40]. The consequence of this is that item scores do not reflect ability and therefore objective comparison between individuals is not possible [41].

## Materials and methods

### Respondents

The dataset comprised 1030 participants, mean age ($M$age) = 32.14 years ($SD$ = 14.02, range of 18 to 71). In total, 465 males ($M$age = 34.28, $SD$ = 14.57, range of 18 to 71) and 565 females ($M$age = 30.37, $SD$ = 13.31, range of 18 to 69) took part. Recruitment occurred via emails to undergraduate and postgraduate UK university students, regional vocational/sports groups, and businesses in the Northwest region of the UK.

### Measures

**Inventory of Personality Organization-Reality Testing Subscale (IPO-RT).** Researchers have often used the IPO-RT [17] to assess inclination to experience reality testing deficits (e.g., [13, 42, 43]). The IPO-RT adopts an information-processing style to belief generation [44]. Commensurate with this approach, the scale assesses "the capacity to differentiate self from non-self, intrapsychic from external stimuli, and to maintain empathy with ordinary social criteria of reality" ([19] p. 120).

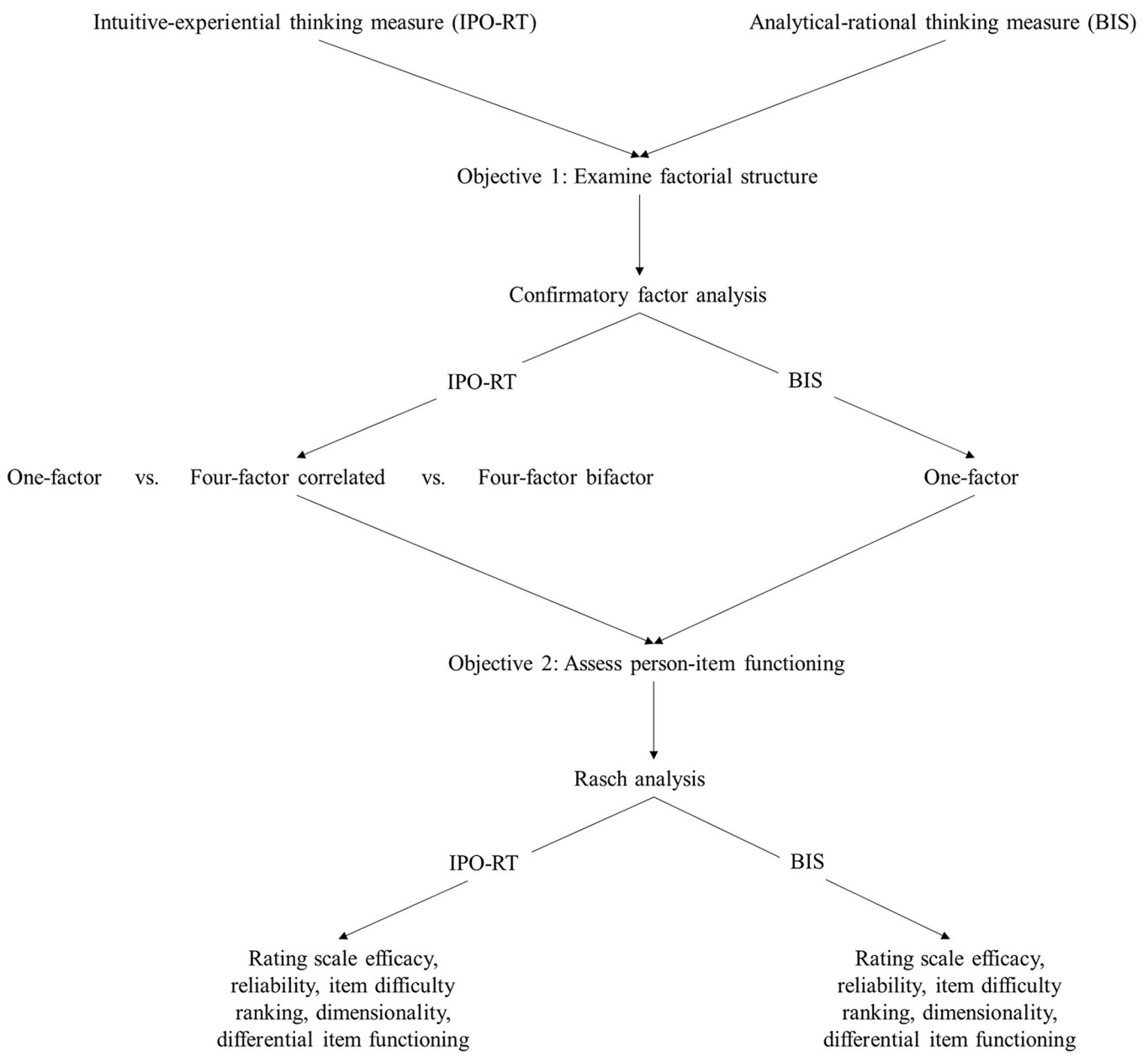

**Fig 1. Flowchart depicting the phases of the research study.**

The IPO-RT is composed of 20-items, which appear as statements (e.g., 'I have seen things which do not exist in reality'). Respondents read each item and signify endorsement on a five-point Likert scale (1 = never true to 5 = always true). Summation of items produces scores that range from 20 to 100. Higher scores signify the tendency to engage with intra-psychic activity and experience reality testing deficits. Across studies the IPO-RT has demonstrated psychometric integrity (i.e., internal consistency, test–retest reliability, and construct validity). Further psychometric evaluation indicates that the IPO-RT is best explained by a bifactor solution consisting of a general reality factor and four subfactors (Auditory and Visual Hallucinations, Delusional Thinking, Social Deficits, and Confusion). Omega reliability and factor loadings

support the notion that the IPO-RT measures a single latent construct. This solution displayed gender and partial age invariance [20].

**Belief in Science Scale (BIS).** BIS [18] is a 10-item measure that assesses faith in the methods and outcomes of science. Respondents indicate the degree to which they endorse statements (e.g., 'All the tasks human beings face are soluble by science') via a 6-point scale (1 = Strongly Disagree to 6 = Strongly Agree). Scores are averaged to produce a response range between 1 to 6, with higher scores signifying greater belief in science. The measure has demonstrated validity and internal consistency [18].

**Procedure.** To access the study, respondents clicked on a web link. This took them to the participation information sheet, which outlined the purpose of the research and the procedure. Only respondents fulfilling the inclusion criteria, who specified informed consent advanced to survey completion. Consent was obtained by requesting participants to tick a box indicating that they agreed to take part. Measures were prefaced by the demographics section (i.e., age, preferred gender, and occupation) followed by the self-report scales. To prevent order effects the order of question blocks (demographics and scales) rotated across respondents. To counter common method variance, general and scale instructions emphasized the uniqueness of each measure. This created psychological disconnection between the scales [45]. Additionally, to reduce social desirability effects the study brief informed respondents to work at their own pace and that there were no correct answers. On completion of the online survey, respondents received a debrief, which reiterated the main points from the participation information sheet. The study gained ethical approval as part of a wider project focusing on anomalous thinking from the Faculty of Health, Psychology and Social Care Ethics Committee at Manchester Metropolitan University. Recruitment occurred in January 2022. Authors had access to an anonymised dataset.

## Analysis

Assessment of the IPO-RT and BIS occurred at the construct, factor, and item level using a combination of analytical techniques; consideration of zero-order correlations followed by confirmatory factor analysis (CFA) and Rasch analysis. CFA compared competing IPO-RT and BIS factorial solutions (see Dagnall et al. [20, 24]). Several indices evaluated data-model fit (i.e., chi-square statistic; Comparative Fit Index, CFI; Standardized Root-Mean-Square Residual, SRMR; and Root-Mean-Square Error of Approximation, RMSEA). CFI > .90, SRMR < .08, and RMSEA < .08 signify acceptable fit, whereas RMSEA < .10 indicates marginal fit [46]. Regarding factor loadings, values > .30 are acceptable [47].

Subsequent measurement information at the person and item level was provided by Rasch analysis via Winsteps software [48]. This can be employed "as a confirmatory test of the extent to which scales have been successfully developed according to explicit a priori measurement criteria" ([49], p. 196). The Rasch Rating Scale Model [33] was utilised. Joint maximum likelihood estimated the parameters for analysis, and five criteria were considered: rating scale efficacy, reliability, item difficulty ranking, dimensionality, and differential item functioning. These criteria are commonly applied when using Rasch analysis (e.g., [50, 51]).

## Results

### Data screening and descriptive statistics

Examination of skewness and kurtosis for the IPO-RT and BIS indicated item statistics were consistent with normal distributions (i.e., skew < 2, kurtosis < 5) [52]. Average total scores (IPO-RT $M$ = 37.37, BIS $M$ = 51.50) aligned with results obtained using non-clinical samples (e.g., [17, 26]). At the total scale (construct) level, the zero-order correlation between IPO-RT

and BIS was moderate ($r$ = -.23, $p$ < .001 [-.28, -.17]) in accordance with the guidelines of Gignac and Szodorai [53]. Higher scores on analytic-rational thinking (BIS) corresponded with lower scores on intuitive-experiential thinking (IPO-RT). Post-hoc analysis using one-way ANOVA examined this relationship at the quartile level of BIS, indicating a main difference between quartiles and IPO-RT, $F(3,1029)$ = 23.17, $p$ < 0.001. No significant difference existed between quartiles 1 and 2 (i.e., below the median), $M_{\text{difference}}$ = 1.21, $p$ = 1.0 [-1.31, 3.74].

Above the median, significant differences existed: quartile 1 vs. 3, $M_{\text{difference}}$ = 3.84, $p$ < .001 [1.36, 6.32]; quartile 1 vs. 4, $M_{\text{difference}}$ = 7.56, $p$ < .001 [4.97, 10.15]; quartile 2 vs. 3, $M_{\text{difference}}$ = 2.62, $p$ = .034 [.12, 5.12]; quartile 2 vs. 4, $M_{\text{difference}}$ = 6.34, $p$ < .001 [3.73, 8.95]; quartile 3 vs. 4, $M_{\text{difference}}$ = 3.72, $p$ < .001 [1.16, 6.27]. Moreover, the correlation between below the median BIS scores and IPO-RT was non-significant, $r$ = -.02, $p$ = .614 [-.10, .06]. The association between above the median BIS scores and IPO-RT was significant, $r$ = -.20, $p$ < .001 [-.28, -.12]. This suggests that a moderate level of BIS was necessary to indicate a relationship with IPO-RT.

## Factor analysis

Consistent with Dagnall et al. [20], CFA tested one-factor, four-factor, and bifactor depictions of the IPO-RT. The one-factor model demonstrated unsatisfactory data-fit across indices (Table 1). In addition, unsatisfactory fit existed for a four-factor correlated solution. The four-factor bifactor model, however, evidenced satisfactory fit. Inspection of modification indices revealed an improvement in model fit if the error term between item 14 and 15 was allowed to correlate. Like Dagnall et al. [20], a model with this amendment was specified.

A general factor accounted for a greater average loading (.52) in comparison with the specific bifactors (Confusion = .45, Auditory and Visual Hallucinations = .10, Delusional Thinking = .26, Social Deficits = .40). Moreover, all items (apart from Item 1 = .23) loaded > .30 on the general factor.

Assessment of bifactor-specific criteria provided further evidence for the dominance of a general factor vs. specific bifactors. Explicitly, Explained Common Variance (ECV) was considered alongside Percentage of Uncontaminated Correlations (PUC) and hierarchical omega ($\omega_h$). ECV > .60, PUC > .70, and $\omega_h$ > .70 were supportive of unidimensionality [54]. Moreover, greater factor determinacy (FD > .90) and construct replicability (H > .80) for the general factor in tandem with higher relative omega vs. specific bifactors, indicated that the measure reflected a general construct. Lastly, Item Explained Common Variance (IECV) inferred how representative an item is of a dimension [55].

**Table 1. Fit indices for the IPO-RT and BIS models.**

| Model | $\chi^2$ | $df$ | CFI | SRMR | RMSEA (90% CI) |
|---|---|---|---|---|---|
| IPO-RT | | | | | |
| One-factor | 2459.92** | 170 | .70 | .08 | .11 (.11-.12) |
| Four-factor | 1565.47** | 164 | .82 | .08 | .09 (.08-.09) |
| Bifactor | 842.50** | 150 | .91 | .05 | .06 (.06-.07) |
| Bifactor with correlated error | 665.27** | 149 | .93 | .05 | .05 (.05-.06) |
| BIS | | | | | |
| One-factor | 504.27** | 34 | .93 | .04 | .11 (.10-.12) |
| One-factor with correlated error | 386.60** | 31 | .95 | .03 | .10 (.09-.11) |

*Note.* **$\chi^2$ significant at $p$ < .001

ECV was .61, with PUC of .76 and $\omega_h$ of .82. High FD (.92), relative omega (.89) and H (.89) existed. Though each bifactor exhibited satisfactory omega (Confusion $\omega$ = .82, Auditory and Visual Hallucinations $\omega$ = .73, Delusional Thinking $\omega$ = .80, Social Deficits $\omega$ = .80), relative omega was low (Confusion = .36, Auditory and Visual Hallucinations = .27, Delusional Thinking = .30, Social Deficits = .35). In addition, 50% of items possessed IECV > .70 (Table 2), with 45% of items evidencing IECV greater than the .80 threshold indicative of unidimensionality [55]. These results suggest that the IPO-RT is more resemblant of a unidimensional instrument, with some variance accounted for by specific bifactors. However, this variance is not sufficient to infer multidimensionality.

Assessment of the factorial composition of the BIS focused on the supported one-factor solution (see [24]). Acceptable fit existed across all indices apart from RMSEA (Table 1). Reasons for high RMSEA were unclear, however, this can often occur with relatively large samples. This is because RMSEA, like chi-square, increases in accordance with sample size [56]. Nonetheless, scrutiny of MI reported an improvement in RMSEA based on correlating error

**Table 2. Factor loadings and Item Explained Common Variance (IECV) for IPO-RT and BIS scales.**

| Scale | Sub-scale | Item | General factor | Bifactor | | | | IECV | Mean IECV |
| --- | --- | --- | --- | --- | --- | --- | --- | --- | --- |
| | | | | CON | AVH | DT | SD | | |
| IPO-RT | CON | 1 | .23 | .49 | | | | .40 | .69 |
| | | 3 | .47 | .65 | | | | .67 | |
| | | 6 | .50 | .23 | | | | 1.00 | |
| | AVH | 2 | .56 | | .05 | | | .64 | .63 |
| | | 5 | .53 | | .03 | | | .99 | |
| | | 7 | .55 | | .66 | | | .23 | |
| | | 8 | .58 | | .65 | | | .35 | |
| | | 9 | .57 | | .42 | | | .85 | |
| | | 16 | .57 | | .41 | | | .70 | |
| | DT | 11 | .70 | | | .01 | | .97 | .67 |
| | | 12 | .48 | | | .62 | | .91 | |
| | | 14 | .53 | | | .18 | | .36 | |
| | | 15 | .59 | | | .09 | | .42 | |
| | | 17 | .55 | | | .15 | | 1.00 | |
| | | 18 | .65 | | | .18 | | .59 | |
| | | 19 | .52 | | | .65 | | .42 | |
| | SD | 4 | .58 | | | | -.01 | .44 | .81 |
| | | 10 | .46 | | | | .56 | .91 | |
| | | 13 | .35 | | | | .68 | 1.00 | |
| | | 20 | .43 | | | | .35 | .90 | |
| BIS | | 1 | .58 | | | | | | |
| | | 2 | .46 | | | | | | |
| | | 3 | .77 | | | | | | |
| | | 4 | .75 | | | | | | |
| | | 5 | .78 | | | | | | |
| | | 6 | .88 | | | | | | |
| | | 7 | .86 | | | | | | |
| | | 8 | .75 | | | | | | |
| | | 9 | .81 | | | | | | |
| | | 10 | .64 | | | | | | |

*Note*. CON = Confusion, AVH = Auditory and Visual Hallucinations, DT = Delusional Thinking, SD = Social Deficits

between items 1, 2, and 10. These were also the items with the lowest factor loadings (Item 1 = .60, Item 2 = .48, Item 10 = .65). Remaining items possessed factor loadings greater than .75. Rerunning the analysis with this correlated error produced a model with improved (and satisfactory) RMSEA. All items (aside from 1, 2, and 10) retained strong factor loadings, and an average loading of .73 existed (Table 2). These findings support unidimensionality of the BIS. The reliability of the scale was good, $\omega$ = .92.

## Rasch analysis

Previous literature (e.g., [20, 24]) and CFA findings supported sufficient unidimensionality of the IPO-RT and BIS. Therefore, these were assessed as unidimensional scales within Rasch analysis. An initial examination of item fit revealed all IPO-RT items possessed adequate Infit and Outfit Mean Squares (MNSQs) (between .5 and 2.0) alongside positive and strong Point-Measure Correlations (PTMEAs) (i.e., > .4). This implies a lack of 'noise' or randomness existed within the measure. However, for the BIS, Item 2 evidenced an Outfit MNSQ of 2.06. Re-running the analysis without Item 2 revealed the remaining items possessed Infit and Outfit MNSQs between .5 and 2.0 alongside PTMEAs > .4 (Table 3). Assessment of the rating scale for the IPO-RT revealed acceptable Infit and Outfit MNSQ below 2.0 (Table 4), suggesting a reasonable usage of response options [57]. Though, the sample were more inclined to disagree with items. Similarly, for the BIS acceptable Infit and Outfit MNSQ existed for the response scale. For this scale, participants were more inclined to endorse items.

Summary fit statistics for the IPO-RT reported a high person reliability of .87, and a Person Separation Index (PSI) of 2.58. Person reliability for the BIS was also high (.91), with a PSI of 3.19. The PSI results infer participants are being effectively separated into more than one ability level in each instance [57]. Item reliability was high for the IPO-RT (.99) and the BIS (1.0).

Wright maps demonstrated the distribution of items in terms of their complexity relative to the sample. For the IPO-RT (Fig 2), items were generally difficult to endorse, as evidenced by an average item difficulty greater than the mean endorsement. Items 16, 4, and 8 were the most difficult to endorse, whereas Item 1 was the easiest to endorse. For the BIS (Fig 2), items were easy to endorse, given mean endorsement was greater than the average item difficulty. Indeed, only Item 7 (the most difficult to endorse) appeared above the mean endorsement level. Items 1 and 10 were the most straightforward to endorse.

Principal Components Analysis of the residuals assessed scale dimensionality. A single dimension for the IPO-RT explained 41.2% of variance. The unexplained variance in the first contrast was 8.5%, with an Eigenvalue of 2.9. This is below the threshold of 3.0, which is suggestive of a component [48]. A single dimension accounted for 68.3% of variance for the BIS. In addition, the unexplained first contrast variance was 1.8%. For both scales, these results support unidimensionality.

Item equivalence was assessed for the IPO-RT and BIS using differential item functioning (DIF) and comparing gender (males vs. females). One IPO-RT item (Item 19) displayed DIF contrast > .50 (Mantel-Haenszel $p$ = .002). Specifically, females found this item harder to endorse than males (DIF contrast of .51). No meaningful DIF existed for the BIS.

## Discussion

The present study provides support for the supposition that the IPO-RT and BIS assess preference for conceptually different aspects of thought. In this context, the inverse relationship is consistent with the central dual processing model assumption. Explicitly, it aligns with previous investigations that have used the IPO-RT and BIS as proxy measures to assess intuitive-experiential and analytical-rational systems. For instance, Dagnall et al. [24] ($r$ = -.28) and

**Table 3. Item fit statistics.**

| Item | Difficulty | Infit MNSQ | Outfit MNSQ | PTMEA Corr. |
|------|-----------|------------|-------------|-------------|
| IPO-RT | | | | |
| Item 1 | -1.40 | 1.08 | 1.20 | .40 |
| Item 2 | -.47 | .86 | .86 | .58 |
| Item 3 | -.75 | .96 | .98 | .54 |
| Item 4 | .89 | 1.13 | .97 | .52 |
| Item 5 | -.71 | .88 | .93 | .54 |
| Item 6 | -.51 | 1.11 | 1.14 | .53 |
| Item 7 | .57 | .97 | .90 | .58 |
| Item 8 | .81 | .97 | .78 | .59 |
| Item 9 | -.03 | .96 | .91 | .61 |
| Item 10 | -.23 | 1.01 | 1.02 | .51 |
| Item 11 | .26 | .80 | .75 | .63 |
| Item 12 | .50 | 1.16 | 1.04 | .52 |
| Item 13 | .10 | 1.18 | 1.21 | .42 |
| Item 14 | .07 | 1.07 | 1.09 | .56 |
| Item 15 | .10 | 1.22 | 1.15 | .59 |
| Item 16 | .80 | 1.19 | 1.01 | .56 |
| Item 17 | -.39 | 1.01 | 1.02 | .57 |
| Item 18 | .18 | .90 | .85 | .62 |
| Item 19 | .49 | 1.14 | .99 | .56 |
| Item 20 | -.28 | 1.11 | 1.12 | .49 |
| BIS | | | | |
| Item 1 | -1.39 | 1.75 | 1.49 | .59 |
| Item 3 | -.08 | 1.03 | .97 | .76 |
| Item 4 | .65 | 1.07 | 1.09 | .77 |
| Item 5 | .63 | .91 | .90 | .80 |
| Item 6 | .26 | .75 | .70 | .82 |
| Item 7 | .86 | .80 | .74 | .83 |
| Item 8 | .45 | 1.08 | 1.12 | .76 |
| Item 9 | -.34 | .83 | .77 | .77 |
| Item 10 | -1.03 | 1.30 | 1.28 | .65 |

*Note*. The MNSQ acceptable limits to productive measurement were 0.5 to 2.0. Values beyond these limits are considered misfitting. Results for BIS are after the removal of Item 2.

Irwin et al. [42] ($r$ = -.32) observed a similar moderate relationship to the one reported in the current paper ($r$ = -.23) (see [53]). However, caution is required since the negative correlation explained only approximately 5% of the observed variance. This finding aligns with the observation of Dagnall et al. [24] that moderate levels of BIS are required to facilitate a decrease in intuitive-experiential thinking.

Moreover, it is difficult to draw meaningful conclusions with other proxy measures of intuitive-experiential and analytical-rational processing/thinking because they vary markedly across the literature. This comparison is further complicated by the fact that measures such as the dimensions of the REI, assess only relatively narrow aspects of construct domain. In contrast, the IPO-RT and BIS are more inclusive, general measures of dual processes.

At a factorial level, strong evidence existed in support of a bifactor structure for IPO-RT. Explicitly, a general reality testing factor accounted for the majority of variance, with some

**Table 4. Rating scale effectiveness.**

| | Category | Count (%age) | Avg. Measure | Infit MNSQ | Outfit MNSQ |
|---|---|---|---|---|---|
| IPO-RT | | | | | |
| | 1 Never true | 8793 (43) | -2.06 | .93 | .97 |
| | 2 Rarely true | 5636 (27) | -1.24 | 1.00 | .79 |
| | 3 Sometimes true | 4086 (20) | -.71 | .95 | .89 |
| | 4 Often true | 1682 (8) | -.26 | 1.11 | 1.17 |
| | 5 Always true | 403 (2) | .08 | 1.48 | 1.63 |
| BIS | | | | | |
| | 1 Strongly disagree | 1009 (11) | -1.65 | 1.08 | 1.31 |
| | 2 Disagree | 904 (10) | -.78 | 1.12 | 1.15 |
| | 3 Slightly disagree | 1188 (13) | -.26 | .89 | .85 |
| | 4 Slightly agree | 1617 (17) | .36 | .93 | .93 |
| | 5 Agree | 2137 (23) | 1.17 | 1.00 | 1.00 |
| | 6 Strongly agree | 2415 (26) | 2.30 | 1.07 | 1.07 |

*Note*. Results for BIS are after the removal of Item 2

variance explained by the specific bifactors. This variance was not sufficient to indicate multi-dimensionality. These findings help to resolve the variations inherent within past research. Particularly, the inference that the IPO-RT is strictly unidimensional [17] or multifactorial [20, 29].

In addition, support for a dominant general factor has implications for the administration of the IPO-RT in practice/research. Specifically, it advocates administration of the scale in its entirety. Though, lack of support for a one-factor solution alongside a degree of non-redundant variance for the bifactors indicates that consideration of specific factors is necessary and advisable. Indeed, these could usefully be applied alongside total scores when evaluating performance. This recommendation aligns with preceding research evidencing support for bifactorial models that reflect a dominant general factor alongside non-redundant bifactors (e.g., [58]).

Rasch analysis of the IPO-RT suggested satisfactory reliability, reasonable evidence of unidimensionality, and a lack of misfitting items. However, DIF existed for item 19 ('I believe that things will happen simply by thinking about them'), with females finding this item more challenging. It is unclear why this occurred, given that gender differences are not claimed to exist with regards to the measure [17]. However, since this is the first study to examine item response functioning within the IPO-RT, it would be useful for subsequent studies to corroborate this finding and ascertain if gender-DIF is an inherent concern relating to this item.

The person-item map indicated that IPO-RT items were generally challenging to endorse for the sample. It is likely that this result occurred because the sample did not exhibit particularly high levels of reality testing deficits ($M = 37.37$) and consequently struggled to relate to the clinically oriented phrasing of some of the items. Indeed, although the IPO-RT was designed for use with clinical and non-clinical samples, its purpose is to assess reality testing as a feature of personality disorganisation/pathology (see [17]).

Congruent with prior research (i.e., [24]), support existed for a one-factor solution for BIS. This finding adds further evidence for the existence of a singular belief in science component underpinning the BIS. At the item level, correlation of error terms among items 1, 2 and 10 improved model fit. Due additionally to weaker factor loadings for these items, it is possible that they represent a discrete component. Interestingly, these items possess similarities in content. Specifically, 1 ('Science provides us with a better understanding of the universe than does

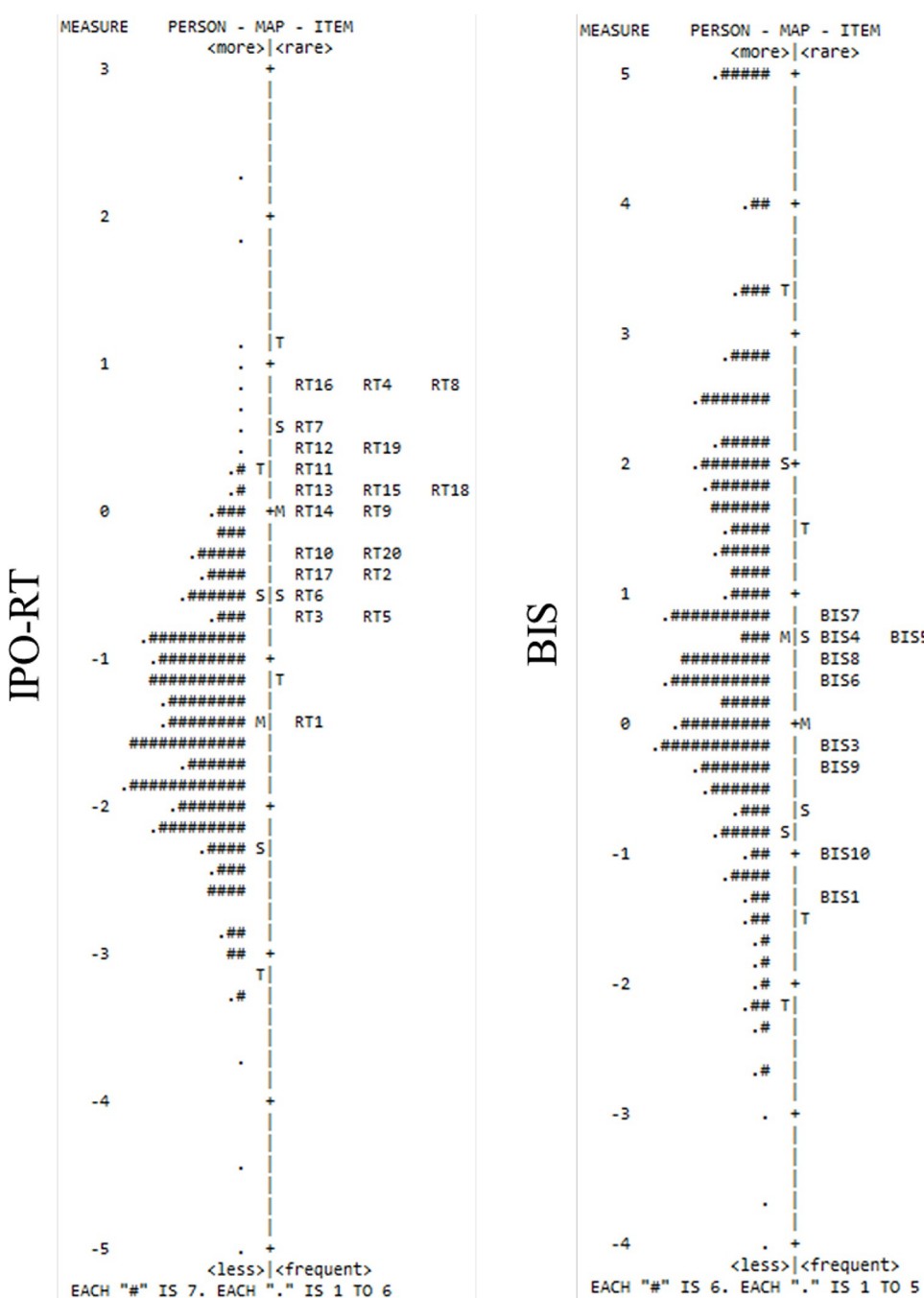

**Fig 2. Person-Item maps of IPO-RT and BIS.** *Note.* M = Mean persons' ability or mean items' difficulty; S = one standard deviation; T = two standard deviations.

religion'), 2 ('In a demon-haunted world, science is a candle in the dark. (Carl Sagan)'), and 3 ('Scientists and science should be given more respect in modern society') centre on science in terms of its relationship with external societal/universe-based features.

It would be useful for future research to examine if these items represent a divergent factor. However, Rasch analysis recommended the removal of item 2 due to high infit statistics. Assessing a factor with less than three items is typically not recommended [59]. Moreover, this

item exhibited deviation to the remaining items, and it is possible that the content of the item may be responsible. Explicitly, the item differs in the sense it comprises a quote. Given that misfit existed relative to Outfit, perhaps unexpected responses occurred due to the differing phraseology of the item [60].

Assessment of the BIS without item 2 demonstrated satisfactory psychometric properties (unidimensionality, reliability, absence of rogue/misfitting items) alongside an absence of gender-DIF. In contrast to the IPO-RT, the person-item map indicated that the sample found the BIS items easy to endorse. This is unsurprising given a relatively high average score existed among the remaining items ($M = 46.27$). Given its relative newness, a lack of norms exists for the BIS; however, the deduction of a high average score in this study is formed based on a total possible score of 60. Therefore, the sample exhibited a relatively high belief in science, which was reflected by a ready endorsement of the items.

## Limitations and conclusions

Though satisfactory construct and psychometric evidence occurred for the IPO-RT and BIS, several limitations exist. A problem with using self-report measures to evaluate metacognitive processes is that they provide only secondary, subjective insights into mental processes. Thus, even when self-report instruments are psychometrically established and provide consistent responses, scores may not accurately predict objective performance [61]. Additionally, appraisal of specific aspects of metacognition is difficult since processes operate in a holistic, integrated manner. The ensuing interactions between the executive processes that direct and manage cognitive activities [62] are difficult to interpret [13].

A further issue with self-evaluations of thinking style is that individuals can report only awareness. Some aspects of processing are automatic and not available to consciousness [63, 64]. These factors explain why studies often report that the relationship between subjective ratings and actual performance is often weak [65, 66]. Although this is a general concern, it may be especially true of the IPO-RT, which requires discernments of spontaneous, abstract mental processes. To establish validity, it is therefore necessary to compare ratings with objective criteria. Indeed, there are several established tests that assess the ability to engage in evaluative processing (e.g., Watson Glaser Critical Thinking Appraisal), which researchers could use. Also, it is necessary to determine which specific aspects of thinking the IPO-RT and BIS index. This is important because the instruments may more effectively index aspects of thought.

Recognising the breadth of intuitive-experiential and analytical-rational processing/thinking, evaluating the performance of IPO-RT and BIS alongside objective tests would enable researchers to determine the validity of scores. In the case of the BIS this is especially vital since the instrument assesses faith in science as opposed to the tendency to engage in rational thought. Although, advocacy of the scientific method and the tendency to engage in logical inquiry typically overlap, belief does not necessarily predict action. Indeed, radical avowal of science without appreciation of its limitations can produce uncritical, dogmatic appraisal of data [67]. True appreciation of complex real-world issues requires methodical, critical, objective, evaluation of information. Uncritical support for science can result in a truncated or biased evaluation of evidence. Certainly, within individuals who hold strong views there is a tendency to seek verification and dismiss alternatives [68]. Thus, it is possible to score high on BIS but still think or process information experientially.

## Acknowledgments

The authors would like to thank the research participants, without whom the study would not be possible.

## Author Contributions

**Conceptualization:** Andrew Denovan, Neil Dagnall.

**Data curation:** Andrew Denovan, Neil Dagnall, Ken Drinkwater.

**Formal analysis:** Andrew Denovan, Neil Dagnall, Ken Drinkwater, Álex Escolà-Gascón.

**Investigation:** Andrew Denovan.

**Methodology:** Andrew Denovan, Álex Escolà-Gascón.

**Project administration:** Ken Drinkwater.

**Validation:** Álex Escolà-Gascón.

**Visualization:** Andrew Denovan, Álex Escolà-Gascón.

**Writing – original draft:** Andrew Denovan, Neil Dagnall.

**Writing – review & editing:** Andrew Denovan, Neil Dagnall, Ken Drinkwater, Álex Escolà-Gascón.

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
