## [Decision Letter · Decision Letter 0]

23 Jul 2024

PONE-D-23-09482The Inventory of Personality Organization-Reality Testing Subscale and Belief in Science Scale: Confirmatory factor and Rasch analysis of thinking style measuresPLOS ONE

Dear Dr. Denovan,

Thank you for submitting your manuscript to PLOS ONE. After careful consideration, we feel that it has merit but does not fully meet PLOS ONE’s publication criteria as it currently stands. Therefore, we invite you to submit a revised version of the manuscript that addresses the points raised during the review process.

We look forward to receiving your revised manuscript.

Kind regards,

Frantisek Sudzina

Academic Editor

PLOS ONE

Reviewers' comments:

Reviewer's Responses to Questions

**Comments to the Author**

1. Is the manuscript technically sound, and do the data support the conclusions?

Reviewer #1: Yes

Reviewer #2: Yes

2. Has the statistical analysis been performed appropriately and rigorously? 

Reviewer #1: Yes

Reviewer #2: Yes

3. Have the authors made all data underlying the findings in their manuscript fully available?

Reviewer #1: Yes

Reviewer #2: Yes

4. Is the manuscript presented in an intelligible fashion and written in standard English?

Reviewer #1: Yes

Reviewer #2: Yes

5. Review Comments to the Author

Reviewer #1: Dear authors,

The manuscript is good enough, it only needs a few additional figures in several parts. For example, a flowchart can be added to the introduction, as well as examples for research instruments, namely the Inventory of Personality Organization-Reality Testing Subscale (IPO-RT) and the Belief in Science Scale (BIS).

Best regards,

Reviewer #2: Very interesting this article.Tthe study reveals adequate methodological capacity. The theoretical framework refers the reader to studies on the variables that are appropriate to the research objectives. The presentation of the data appears to be consistent, based on the methodological and ethical definition. The discussion and conclusions also appear to be consistent.

-Proposal to improve bibliographical references in accordance with APA standards, for example reference nº66

6. PLOS authors have the option to publish the peer review history of their article (what does this mean?). If published, this will include your full peer review and any attached files.

Reviewer #1: No

Reviewer #2: No

---

## [Author Response · Author response to Decision Letter 0]

19 Aug 2024

Comments to the Author

Review Comments to the Author

Reviewer #1: Dear authors,

The manuscript is good enough, it only needs a few additional figures in several parts. For example, a flowchart can be added to the introduction, as well as examples for research instruments, namely the Inventory of Personality Organization-Reality Testing Subscale (IPO-RT) and the Belief in Science Scale (BIS).

Best regards,

Response:

Thank you for your feedback. We have included a figure with a flowchart of the research phases. Please note that we originally included examples of items from the IPO-RT and BIS. In terms of your other points, we have included a recommendation in the Abstract, and stated the research objectives more explicitly. Please note that we mentioned in the original manuscript when the research was conducted.

Reviewer #2: Very interesting this article.Tthe study reveals adequate methodological capacity. The theoretical framework refers the reader to studies on the variables that are appropriate to the research objectives. The presentation of the data appears to be consistent, based on the methodological and ethical definition. The discussion and conclusions also appear to be consistent.

-Proposal to improve bibliographical references in accordance with APA standards, for example reference nº66

Response:

Thank you for your supportive review of the manuscript. Please note that we followed the PLOS One referencing style requirements for the manuscript (Vancouver).

---

## [Editor Report · Decision Letter 1]

23 Aug 2024

The Inventory of Personality Organization-Reality Testing Subscale and Belief in Science Scale: Confirmatory factor and Rasch analysis of thinking style measures

PONE-D-23-09482R1

Dear Dr. Denovan,

We’re pleased to inform you that your manuscript has been judged scientifically suitable for publication and will be formally accepted for publication once it meets all outstanding technical requirements.

Kind regards,

Frantisek Sudzina

Academic Editor

PLOS ONE
---

## [Editor Report · Acceptance letter]

28 Aug 2024

PONE-D-23-09482R1 

PLOS ONE

Dear Dr. Denovan, 

I'm pleased to inform you that your manuscript has been deemed suitable for publication in PLOS ONE. Congratulations! Your manuscript is now being handed over to our production team.

Kind regards, 

on behalf of

Dr. Frantisek Sudzina 

Academic Editor

PLOS ONE